# Age-Related Macular Degeneration: Role of Oxidative Stress and Blood Vessels

**DOI:** 10.3390/ijms22031296

**Published:** 2021-01-28

**Authors:** Yue Ruan, Subao Jiang, Adrian Gericke

**Affiliations:** Department of Ophthalmology, University Medical Center, Johannes Gutenberg University Mainz, Langenbeckstr. 1, 55131 Mainz, Germany; sjiang@uni-mainz.de

**Keywords:** age-related macular degeneration, pathogenesis, oxidative stress, dysregulated lipid metabolism, choroidal vascular dysfunction, genetic factor

## Abstract

Age-related macular degeneration (AMD) is a common irreversible ocular disease characterized by vision impairment among older people. Many risk factors are related to AMD and interact with each other in its pathogenesis. Notably, oxidative stress and choroidal vascular dysfunction were suggested to be critically involved in AMD pathogenesis. In this review, we give an overview on the factors contributing to the pathophysiology of this multifactorial disease and discuss the role of reactive oxygen species and vascular function in more detail. Moreover, we give an overview on therapeutic strategies for patients suffering from AMD.

## 1. Introduction

Age-related macular degeneration (AMD) is a common irreversible sight-threatening disease characterized by progressive degeneration of the central retina, preferentially involving the retinal photoreceptors, the retinal pigment epithelium (RPE), the Bruch’s membrane (BM), or the choroidal microcirculation in the macular region [1]. In 2020, AMD accounted for 5.4% (1.8 million) of blindness among the global 33.6 million blind adults over the age of 50 and is the fourth leading cause of blindness worldwide [2]. Moreover, the prevalence of AMD is predicted to rise continuously and rapidly based on the increasing average life expectancy [3,4]. It has been estimated that in 2040, 288 million people will be affected by AMD worldwide [5]. Depending on the degree of disease severity, patients perceive different decreases in their quality of life [6]. For example, patients with mild AMD perceive a 17% decrement in their quality of life, while people with severe AMD report on a 63% reduction in the quality of life [6]. In addition to the deleterious effects on patients’ quality of life, treatment of AMD causes high economic costs [6]. The annual loss of gross domestic product due to AMD was $2.6 billion in Canada in 2005, while the annual loss was approximately $4.6 billion in the United States in 2016 [1,6].

The first reports of the pathogenetic process underlying AMD were described by Donders et al. in 1855 and by Nagel et al. in 1868 [7,8]. By 1965, the terminology of AMD was becoming more and more accepted [9]. AMD is classified into three clinical stages: early, intermediate, and advanced AMD [10]. The presence of drusen (>63 and ≤125 μm in diameter) is the earliest clinical feature of early AMD, which impairs the patients’ ability of dark adaptation during the transition from high to low illumination environments [10]. Most central visual loss occurs in the intermediate and advanced stages of AMD. Advanced AMD includes two categories: geographic atrophy (GA) and neovascular AMD [11]. GA is characterized by slowly progressive deterioration of the RPE, photoreceptor layer, and choroidal capillaries in the macula, leading to progressive vision loss over several years [12]. Neovascular AMD, also known as exudative AMD, is characterized by the invasion of new immature choroidal vessels breaking through the BM into the retina, causing exudates, hemorrhages and detachment of the RPE or retina. This disease form causes more rapid progressive loss of vision than GA [12,13].

Many risk factors are related to AMD and interact with each other in its pathogenesis, making AMD a complex multifactorial disease [14]. The multifactorial etiology includes demographic factors (age, gender, and ethnicity), epidemiological risk factors (body-mass index, smoking, diet, and gene polymorphisms, e.g., mutations in the complement cascade), and environmental risk factors (exposure to sunlight and to chemical substances) [15,16,17,18]. Notably, oxidative stress and choroidal vascular dysfunction have been suggested to be the most important trigger factors of AMD pathogenesis [19,20,21]. Oxidative stress plays an important role in aging diseases including AMD, especially owing to the much higher oxygen consumption by the retina than by any other tissue [22]. In addition, the proliferative abnormal choroidal vasculature grows into the subretinal space in the exudative form of AMD, eventually causing detachment of the RPE and consecutively vision loss [23]. It is more and more recognized that genetic risk factors have a critical relevance to the oxidative stress response and choroidal vascular dysfunction in AMD [24,25]. However, still little is known regarding the interplay between these two pathophysiological factors and their link to genetic factors in AMD pathogenesis. As yet, up to 90% of AMD patients worldwide are still untreatable [13]. This situation makes it an urgent priority to better understand the pathophysiology of AMD and to design targeted therapies for this disease.

In this review, we focus on the current understanding of AMD pathogenesis, especially on the role of oxidative stress and choroidal vascular dysfunction. We also provide an overview on the interrelation between different pathogenic factors in AMD pathophysiology and present potential therapeutic approaches.

## 2. The Macula

The macula is an oval-shaped spot localized in the center of the retina, which is responsible for clear and fine detail vision [26]. The macula has a diameter of about 5 mm and can be subdivided into six areas: the umbo, foveola, foveal avascular zone, fovea, parafovea, and perifovea areas [26]. The fovea is located at the center of the macula and contains the largest concentration of cones in the retina, enabling high-resolution vision [26]. The central region of the macula, which is 250 to 600 μm in diameter and termed the foveal avascular zone, lacks retinal blood vessels and is supplied by the choroidal circulation [27].

The fovea is composed of few layers from anterior to posterior: an extremely thin inner plexiform layer, the outer nuclear layer, the cones, and the RPE layer [28]. The foveola lies in the center of the fovea and contains only cone photoreceptors and unique Müller cells with optical fiber characteristics [29]. Moreover, peripheral areas of the macula and the rest of the retina contain both rod and cone cells [30]. The RPE is attached to the cone photoreceptors and carries out many functions including phagocytosis of the photoreceptors’ outer segment membrane, maintenance of the physiological functions of the choriocapillaris, conversion and storage of retinoid, absorption of scattered light, and transport of ions and fluid [31]. RPE cells are taller in the fovea than in non-foveal areas [32]. The BM is attached to the basal surface of the RPE, an elastic semi-permeable barrier for major metabolic transport and exchange [33]. Adjacent to BM, the choriocapillaris is composed of fenestrated capillaries in the innermost layer of the choroid that provides blood supply to the RPE and the macula (Figure 1) [34]. A better understanding of macular anatomy can significantly improve our understanding regarding the role of risk factors in the pathogenesis of AMD.

## 3. Clinical Classification of AMD

AMD is a degenerative disease of the retina, which leads to changes in photoreceptors, RPE, BM, and/or choriocapillaris, eventually resulting in central visual impairment [35,36]. The pathology of AMD is characterized by macular drusen, RPE atrophy, choroidal neovascularization, neurosensory retina detachments, and disciform scars or lesions [37]. According to the clinical manifestation, several classification scales of AMD have been developed. For example, The Age-Related Eye Disease Study (AREDS) research group divided patients into four AMD categories, depending on the size and extent of drusen, presence of GA, and neovascular changes [38]. Later, the AREDS research group developed a nine-step fundus photographic severity scale for AMD, combining the six-step drusen area and five-step pigmentary abnormality area scales for tracking the progression of AMD and providing baseline risk categories [37]. However, the AREDS nine-step severity scale is overly complicated and not useful for clinical work [39]. Consequently, the AREDS research group proposed a five-step simplified clinical scale for AMD, which is clinically more relevant [40].

However, the precise definition for clinical classification of AMD is still under discussion among clinicians. To deal with this situation, the Beckman Initiative for Macular Research Classification Committee proposed a new clinical classification scheme for AMD in 2013. The Beckman AMD classification system provides a simplified and unified guidance for broad clinical phenotypes by using a modified Delphi technique (Table 1) [41]. Based on the Beckman AMD classification system, the disease is classified into early-stage AMD, intermediate-stage AMD, and late-stage AMD (GA and neovascular AMD) [41]. Early-stage AMD is characterized by the presence of medium-sized drusen (>63 and ≤125 μm) without any impairment of visual function [41]. Intermediate-stage AMD is defined as the presence of large drusen (>125 μm) or/and abnormalities in the RPE [41]. Late-stage AMD (advanced AMD) is classified into two clinical forms: GA (dry or non-exudative AMD) and neovascular AMD (wet or exudative AMD) [41]. GA is defined by the irreversible loss of the RPE and photoreceptor cells, leading to a decrease in visual function [41]. Neovascular AMD is characterized by the invasion of newly fragile choroidal blood vessels growing from the choroid into the retina [41]. This process is known as choroidal neovascularization (CNV), which goes along with blood and fluid leakage, leading to detachment of the retina or RPE and rapid vision loss [41]. Figure 2 describes the clinical manifestation and pathology of AMD from early to late stage.

## 4. Pathogenesis of AMD

### 4.1. Oxidative Stress and AMD

#### 4.1.1. The Macula—An Ideal Environment for the Generation of ROS

It is well known that the retina is one of the highest oxygen-consuming tissues in the human body, utilizing even more oxygen per weight than the brain [42]. The local oxygen metabolic environment in the retina plays an essential role in keeping retinal homeostasis between the supply and consumption of retinal oxygen [42]. The retina continuously transforms light into vision, requiring a marked amount of energy and generating reactive oxygen species (ROS), such as the superoxide (O_2_^•−^), the hydroxyl radical (^•^OH), hydrogen peroxide (H_2_O_2_), and singlet oxygen (^1^O_2_) as normal metabolic byproducts [42]. Generally, ROS are produced during oxidative metabolism under physiological conditions and participate in normal cellular metabolism [43]. However, when the generation of ROS exceeds the capacity of the antioxidant systems, ROS disrupt the balance of redox homeostasis and cause oxidative stress [22].

Owing to the property of high oxygen-metabolism, retinal tissue generates significant amounts of ROS, which makes the retina susceptible to oxidative damage [22]. Evidence shows that the choroidal circulation can only supply the outer retina, while the inner retina is nourished by the retinal vasculature [44]. Importantly, the central 250–600 µm of the macula are devoid of retinal blood vessels and receive blood supply from the underlying choriocapillaris only [33]. Consequently, the RPE is exposed to high ambient oxygen partial pressures of 70–90 mm Hg, which provides an ideal environment for the generation of abundant exogenous ROS [45]. Moreover, the RPE constitutes the outer blood-retinal barrier (oBRB), maintains phagocytosis of around 30,000 photoreceptor outer segments, heat exchange, and vitamin A metabolism, which all produce high levels of ROS [46]. Furthermore, the macula is constantly exposed to light and absorbs light to optimize vision, which causes photo-oxidative stress as an additional source of exogenous oxidative stress [33,47]. An in vitro study showed that short-wavelength light induces ROS production in the mitochondria [48]. The characteristics of the unique sources of retinal ROS generation and high oxygen consumption suggest that oxidative damage is an essential factor in the mechanism of AMD development.

#### 4.1.2. Animal Models of AMD

Experimental models of AMD have been developed in different species, such as mice, rats, rabbits and pigs to better understand the pathogenesis of the disease and to provide suitable preclinical models for drug intervention [49]. These animal models include natural and genetically engineered animal models [50]. A light-induced model is provided for AMD to imply the relationship between oxidative damage and AMD, since light is a natural risk factor involved in AMD, and a light-induced model is easy to produce by varying light intensity and duration [51,52,53,54]. Moreover, transgenic animal models have been studied in AMD over the last few years, such as the nuclear factor-erythroid 2-related factor 2 (*Nrf2*) knock-out (KO) mouse model, the peroxisome proliferator-activated receptor gamma coactivator-1 alpha (*PGC-1α*) KO mouse model, the superoxide dismutase-2 (*SOD2*) KO mouse model and the peroxisome proliferator-activated receptor-β (*PPARβ*) KO mouse model [55,56,57]. These genetic animal models can develop various forms of AMD and enhance the understanding of the disease, offering great possibilities for gene therapeutic approaches.

#### 4.1.3. The Generation of ROS Due to Light Exposure

It is well established that light exposure has the potential to cause detrimental effects in many organs and tissues, such as the skin, cornea, conjunctiva, lens, as well as the RPE and retina [58,59,60]. Large amounts of ROS are produced by exposure to ultraviolet light wavelengths from 100 nm to 400 nm and to blue light wavelengths from 400 nm to 500 nm [58,59,60]. It should be noted that the photoreceptors in the macula are directly exposed to light and are absorbing parts of the light spectrum through rhodopsin, a photoreceptor molecule in rods [61]. Since the cornea, anterior chamber, and crystalline lens effectively filter different parts of the ultraviolet spectrum, only a low portion (1% or less) of the ultraviolet band (315–400 nm) reaches the retina [62]. Some studies demonstrated that absorption of ultraviolet rays by the retina results in photochemical reactions via a type 1 mechanism (direct reactions involving proton or electron transfers) and a type 2 mechanism (reactions involving ROS) [60,62]. Likewise, blue light was shown to be capable of inducing substantial ROS formation in the retina and RPE. The generation of ROS during photooxidative stress may damage cellular components (lipids, proteins and deoxyribonucleic acid (DNA)) and, thereby, is responsible for a large part of cytotoxicity [63].

Based on an in vitro study on an organotypic culture system for mouse retinas by Roehlecke and Schumann, it has been proposed that the generation of ROS occurs directly in outer segments of photoreceptors by nicotinamide adenine dinucleotide phosphate oxidase (NOX) as well as by the mitochondria-like activity of the outer segments after visible blue light (405 nm) irradiation with an output power of 1 mW/cm^2^ [63]. In addition, the authors found that blue light rapidly induced ROS generation in photoreceptors of retinal explants after 0.5–1 h by increasing NOX activity (especially NOX2 and NOX4) and demonstrated that a cross-talk between NOX and mitochondria-like activity may stimulate NOX activation [63]. Another study conducted on the human RPE cell line, adult retinal pigment epithelial cell line-19 (ARPE-19), revealed that the mitochondrial electron transport chain is an important source of ROS and that mitochondria-derived ROS played a critical role in the death of cells exposed to short-wavelength blue light (425 ± 20 nm) [48]. Meanwhile, several pigments in the retina, such as rhodopsin, lipofuscin and melanin were shown to be involved in the process of inducing oxidative stress [64,65,66]. Grimm et al. reported that rhodopsin mediated blue-light-induced damage in the retina, which occurred after short time exposure to blue light but not to green light [64]. In the RPE, lipofuscin is derived primarily from phagocytosis of shed photoreceptor outer segments and is considered a heterogeneous waste material that accumulates with age in active postmitotic cells, such as those of the RPE [67,68]. Evidence also implicated that lipofuscin serves as a photoinducible generator of ROS and is an initiator of blue light damage in the RPE [31]. In an in vitro study in cultured RPE cells, Shamsi et al. demonstrated that lipofuscin is capable of inducing lipid peroxidation and reducing the efficacy of the lysosomal and antioxidant enzyme systems in RPE cells [69]. In an ARPE-19 cell culture model constructed by Sparrow et al., the lipofuscin fluorophore, *N*-retinyl-*N*-retinylidene ethanolamine (A2E), contributed to blue light-induced damage in RPE cells [65]. Furthermore, lipofuscin as a potent generator of ROS may exert deleterious effects to the retina by light exposure [70].

#### 4.1.4. Generation of ROS by Dysregulated Autophagy

Autophagy is an intracellular protein degradation process that is widely existent in eukaryotic cells and is essential for cellular homeostasis [71]. Evidence has been provided that the activity of the autophagy pathway is cyclically engaged in the RPE by examining the conversion of microtubule-associated light chain 3 (LC3) to its lipidated form (LC3-II) [72]. The physiological functions of RPE cells are responsible for the phagocytosis of shed photoreceptor outer segments (POSs), which contain large quantities of unsaturated fatty acids [73]. The process of POS phagocytosis requires high oxygen consumption, and large amounts of ROS are generated by NOX or peroxidase in the phagocytic bodies via oxidizing these fatty acids in POSs [74]. A study by Mitter et al. revealed that autophagy plays a pivotal role in protection of the RPE from oxidative stress [75]. Impaired autophagy in the RPE, characterized by an impaired removal of oxidatively damaged proteins and organelles, may exacerbate oxidative stress and contribute to the pathogenesis of AMD [75]. Recent evidence shows that dysfunctional autophagy/mitophagy in the RPE may lead to mitochondrial disintegration by affecting the mitochondrial fission/fusion ratio, resulting in excessive amounts of ROS [76]. Cano et al. reported that cigarette smoke extract, a complex oxidant, inhibited proteasome activity in RPE cells, but as part of a cytoprotective response, upregulated a series of antioxidant and autophagy-related genes, including the scaffolding adaptor protein, p62 [77]. Wang et al. suggested that p62 protects the RPE by facilitating autophagy and by activating an antioxidant response mediated by Nrf2 [78]. Based on their findings, the authors suggested that p62 may be a potential target for AMD treatment [78].

#### 4.1.5. Oxidative Stress and Disease Development

RPE cell death and the resultant dysfunction of photoreceptors are characteristics of late-stage AMD, especially GA [79]. It has been hypothesized that the RPE plays a critical role in the pathogenesis of AMD due to its function as a conduit for metabolic products between the retina and the choriocapillaris, and its central location between photoreceptors and Bruch’s membrane [80]. Since RPE dysfunction induced by oxidative stress is critically involved in AMD pathogenesis, an increasing number of studies is focusing on the mechanisms underlying RPE cell death triggered by oxidative stress [81]. Elevated ROS levels were shown to promote oxidative damage to proteins, lipids, and mitochondrial DNA (mtDNA) [82]. For example, an in vitro study has shown that RPE cells treated with H_2_O_2_ displayed a significantly increased mtDNA damage rate, suggesting that mitochondrial dysfunction is correlated with mtDNA damage [83]. Mitochondrial damage can initiate cell death via the release of mitochondrial proteins, such as cytochrome c, to the cytoplasm through mitochondrial outer membrane permeabilization by the proapoptotic Bcl-2 family members, Bax and Bak [84].

Several researches previously indicated that low ROS levels can cause RPE cell apoptosis, whereas high ROS concentrations may trigger RPE cell necrosis [85]. RPE cells exposed to 50–200 µM of H_2_O_2_ caused mtDNA damage and further promoted apoptosis in vitro [86]. Other studies suggested that mitochondrial oxidative damage in RPE cells is a trigger for RPE and photoreceptor dysfunction in AMD by disrupting metabolism between the retina and the RPE [87,88]. In an in vivo study, Brown et al. disrupted the gene coding for the mitochondrial antioxidant enzyme, SOD2, specifically in the RPE of albino BALB/cJ mice by using an RPE-specific *Cre* expression [87]. The study revealed that the lack of SOD2 was associated with elevated oxidative stress in the RPE, causing RPE and photoreceptor dysfunction [87]. Oxidative stress-induced mitochondrial dysfunction was shown to play a pivotal role in a series of molecular events culminating in activating an intrinsic apoptotic cell death pathway involving caspases-3, -6 and -7 [89]. However, based on studies in human ARPE-19 cells treated with H_2_O_2_ or *tert*-butyl hydroperoxide (tBHP), Hanus et al. concluded that necrosis rather than apoptosis is a major type of cell death in RPE cells in response to oxidative stress [79]. The mechanism of necrosis underlying RPE cell death is characterized by adenosine triphosphate (ATP) depletion, inflammation, and rupture of the nuclear and cytoplasmic membrane, which is mediated by receptor-interacting protein kinase 3 (RIPK3) [79]. Li et al. revealed that H_2_O_2_-induced necrosis of ARPE-19 cells is mediated by cellular calcium overload [85].

ROS-induced RPE cell death can trigger inflammasome activation with defunct proteasomes and autophagy in RPE cells in AMD [90]. Enhanced generation of ROS was shown to cause permeabilization of lysosomal membranes, further activating nod-like receptor family pyrin domain containing 3 (NLRP3) inflammasome in ARPE-19 cells [90]. Another study in cultured ARPE-19 cells has shown that excessive generation of ROS triggered mitogen-activated protein kinases (MAPKs) and the nuclear factor-κB (NF-κB) signaling pathway subsequently activating the NLRP3 inflammasome [91]. The NLRP3 inflammasome is an important mediator of cytokine secretion by connecting to the caspase-1 enzyme via the adaptor protein, apoptosis-associated speck-like protein, which results in caspase-1-mediated release of the pro-inflammatory cytokines, interleukin (IL)-1β, IL-18 and tumor necrosis factor (TNF) [92]. Tseng et al. investigated the effects of NLRP3 inflammasome activation in RPE cells and the mechanisms between inflammasome activation and the pathogenesis of AMD [93]. In the study, destabilization of RPE lysosomes induced NLRP3 inflammasome activation, which may contribute to AMD pathology by generation of IL-1β and caspase-1-mediated cell death termed pyroptosis [93].

Cholesterol and triglycerides are important sources of lipid metabolism in the retina [94]. An active reverse cholesterol transport (RCT) system has been described in the RPE [95,96,97,98]. The lipid-rich retinal photoreceptor outer segments are continuously regenerated and recycled by the RPE, maintaining the function of photoreceptors [94]. The RPE cells recycle 30,000 lipid-rich photoreceptor outer segments back to the photoreceptors daily to preserve visual function [99]. The RPE is centrally involved in lipid metabolism governed by apolipoproteins (apo) in the ocular system [100]. Along with aging, heterogeneous age-related deposits (basal deposits) occur in the BM and are located in the outer collagenous layer [101]. Macrophages are important scavenger cells to remove debris from the BM through phagosomes and lysosomes [94]. In addition, an in vitro study showed that a delay in lipid degradation by lysosomes leads to accumulation of undigested phospholipids in cultured RPE cells [102]. Aging and genetic variants are associated with a decreased efficiency in lysosomal processing and dysregulated lipid metabolism, and result in the accumulation of incompletely digested phospholipids contributing to the development of AMD [103]. Another hypothesis is that serum cholesterol is related to an ocular production of lipoproteins, which is the source of lipid accumulation in the retina [104]. In a retrospective population-based study, it has been found that high intake of saturated fat and cholesterol was associated with an increased risk for early AMD [105]. In 2004, an in vivo study reported that hyperlipidemia due to a high-fat diet caused sub-RPE deposits in the RPE in C57BL/6 mice by altering hepatic and/or RPE lipid metabolism [104].

Excess accumulation of lipid in the RPE and BM may trigger lipid oxidation in the ROS-enriched environment of the macula [106]. The process of lipid peroxidation is possible via two pathways: nonenzymatic phospholipid autoxidation (iron-dependent lipid peroxidation) and enzymatic peroxidation [107]. The enzymatic peroxidation pathway is catalyzed by lipoxygenase, which transforms PUFAs to lipid hydroperoxide molecules (LOOHs) [108]. This process can switch to a non-enzymatic lipid peroxidation process, which results in generation of LOO^•^ radicals [108]. A study in a murine laser-induced CNV model suggested that lysyl oxidase and lysyl oxidase-like 2 may play a significant role in the pathogenesis of AMD, because targeting of both enzymes by antibodies reduced angiogenesis and inflammation as well as fibrosis [109]. The findings by Othman et al. suggested that activation of 12/15-lipoxygenase causes dysfunction of the retinal endothelial cell barrier resulting in increased vascular permeability via involvement of NOX activating the vascular endothelial growth factor receptor 2 (VEGFR2) signaling pathway [110]. An inhibitor of 5-lipoxygenase, pigment epithelium-derived factor receptor, blocked RPE cell death pathways induced by oxidative stress [111]. Lipid peroxidation activates redox-sensitive transcription factors, such as NF-κB, which stimulate the expression of an array of inflammatory cytokines, leading to inflammation that eventually contributes to AMD progression [112].

#### 4.1.6. Genetics Involved in AMD

Due to the multifactorial pathogenesis of AMD, genetic factors are more and more considered as a mechanism in the pathophysiology of the disease. As we mentioned above, mtDNA damage in RPE cells may play a key pathogenetic role in the onset and progression of AMD. Specific genetic variations are preferentially associated with mtDNA damage related to oxidative stress, which influences susceptibility to AMD [113]. For example, recent genetic studies have shown a significant correlation between mtDNA haplogroups and AMD risk factors [114]. Previous work has studied the mtDNA polymorphism, A4917G, a non-synonymous mtDNA single nucleotide polymorphism closely linked to haplogroup T, which independently leads to the development of AMD [115]. The Y402H variant in the complement factor H (*CFH*) has been strongly associated with AMD in the UK population [116]. Ferrington et al. performed genotype analyses for ten common AMD-associated nuclear risk alleles and mtDNA haplogroups and found a significant association between the *CFH* high-risk allele and accelerated mtDNA damage [113]. According to large genetic studies, strong evidence emerged that more genetic variants are also associated with AMD, such as *CFH*-related proteins (*CFHR1* and *CFHR3*), complement component 3 (*C3*), complement factor I (*CFI*), complement component 2 (*C2*), complement factor B (*CFB*), and complement component 9 (*C9*) [117,118,119,120]. In a cohort study of 530 non-familial AMD patients, Fritsche et al. reported in 2010 that deficiency of *CFHR1* and *CFHR3* may have protective effects against the progression of AMD through enhancing local regulation by factor H [121]. In 2016, Fritsche et al. assessed common and rare variations by analyzing by 12 million variants in 43,566 unrelated subjects of predominantly European ancestry. They revealed 34 loci and genes with a rare variant of advanced AMD and 34 loci include genes with compelling biology like the matrix metalloproteinase gene (*MMP9*), the ATP-binding cassette transporter gene (*ABCA1*), and the vitronectin (*VTN*) gene [120]. In addition, they for the first time described three rare variants in/or near *CFH* genes, rs148553336, rs191281603 and rs35292876 [120]. Hence, novel therapies targeting mtDNA polymorphisms, e.g., *CFH*, may become effective for protecting mtDNA in AMD patients with genetic risk factors.

Moreover, genetic studies demonstrated that the apolipoprotein E (*APOE*) gene polymorphism is strongly associated with AMD susceptibility [122]. APOE is the major apolipoprotein of the CNS and an important regulator of cholesterol and lipid transport [122,123]. A population-based study in the Netherlands reported that APOE was expressed in soft drusen and basal laminar deposits in the macula of patients with AMD, suggesting that *APOE* is a susceptibility gene for AMD [122]. In APOE-deficient mice, basal deposits in the BM that resemble alterations observed in aging human eyes, accumulated at an earlier age compared to controls [124]. Zadeh et al. demonstrated that APOE deficiency in mice induced ROS generation and endothelial dysfunction in retinal blood vessels via involvement of the lectin-like oxidized low-density lipoprotein receptor-1 (LOX-1) and NOX2 [125]. APOE is encoded by a gene represented by three alleles: *APOE2*, *APOE3*, and *APOE4*. Increased and decreased risks of AMD are associated with *APOE2* and *APOE4*, respectively [99,102,103]. In support of this concept, Malek et al. concluded that the APOE4 genotype confers an increased risk for AMD in a mouse model [126]. *APOE* is not the only gene associated with AMD susceptibility. A study that executed a genome-wide association scan for AMD in 2157 patients and in 1150 controls indicated that genetic variants near the metalloproteinase inhibitor 3 gene (*TIMP3*) and high-density lipoprotein-associated loci were associated with susceptibility to AMD [127]. In 2010, a genome-wide association study by Neale et al. employing 979 advanced AMD cases and 1709 controls found an association between AMD and a variant in the hepatic lipase gene (*LIPC*), a gene located on chromosome 15q22, in the high-density lipoprotein cholesterol (HDL) pathway [128]. Low-density lipoprotein (LDL) receptor deficient mice exhibited an accumulation of lipid particles in the BM and increased vascular endothelial growth factor (VEGF) levels in the outer retinal layers [129]. Several other genes involved in lipid metabolism were also found to be associated with AMD, such as *C3*, age-related maculopathy susceptibility 2 (*ARMS2*), and scavenger receptor class B member 1 (*SCARB1*) [130]. These gene polymorphisms provide strong support for genetic therapeutic targets for AMD patients.

### 4.2. Choroidal Vascular Dysfunction and AMD

#### 4.2.1. Choroidal Vascular Changes in AMD

In AMD, several changes in the choroidal vasculature have been reported, and these changes were found to vary depending on the disease stage. Measurement and detection of changes in the macular choroidal vasculature can be achieved via various imaging modalities, including fundus autofluorescence (FAF) imaging, optical coherence tomography (OCT), AOCT angiography (OCTA), etc. [131,132,133,134] According to OCTA findings, vessel density in the choriocapillaris was generally reduced in eyes with GA [135]. The OCT angiogram demonstrated that hypoperfusion occurred in the choriocapillaris underlying the area of GA [135].

Among the cases with AMD, approximately 10–15% have neovascular AMD [136]. Neovascular AMD is characterized by abnormal vascular morphology and growth from the choroidal vasculature [137]. Newly fragile choroidal blood vessels grow from the choroid through the BM to the sub-neurosensory retina and the sub-RPE, which is accompanied by exudation and acute visual impairment [138]. The neovascular lesion is classified as type 1, type 2, and type 3 CNV [139]. Type 1 CNV is originating in the choroid as a neovascular complex between the RPE and BM observed by OCTA [120]. Type 2 CNV grows from the choroidal vasculature and passes through the RPE into the subretinal space [120]. Type 3 CNV develops in the neurosensory retina and progresses posteriorly into the subretinal space, clinically seen as tiny intra- and subretinal hemorrhages [120]. Furthermore, a reduced density of vessels in the choriocapillaris was observed near the CNVs in the absence of GA [140]. Identification of the CNV types and observation of the changes in the choroidal vasculature are important for therapeutic indications in AMD patients.

#### 4.2.2. The Mechanism Underlying CNV in AMD

One of the critical targets in the pathogenesis of choroidal vascular dysfunction is VEGF that also plays a role in the pathogenesis of choroidal neovascularization [141]. An in vivo study in 1997 measured VEGF expression in laser-induced CNV in rats and showed that the upregulation of VEGF expression induced CNV [142]. The study also reported that macrophages may be one of the main sources of VEGF in the early stage of the disease [142]. In sub-foveal fibrovascular membranes, VEGF expression was shown to be concentrated in cells resembling fibroblasts, implicating a role of fibroblasts of presumable choroidal origin in the progression of CNV [143]. In addition, Blaauwgeers et al. demonstrated that VEGF-A was produced by differentiated human RPE cells and might be involved in paracrine signaling between the RPE and the choriocapillaris [144]. In 2000, Spilsbury et al. have adopted a recombinant adenovirus vector containing rat VEGF164 cDNA to investigate whether short-term in vivo overexpression of VEGF in RPE cells was sufficient to cause CNV [145]. The authors found that the severity and extent of choroidal neovascularization were influenced by controlling the amount of virus delivered to the subretinal space [145]. Moreover, the results showed that even temporary overexpression of VEGF in RPE cells is sufficient to induce CNV in the rat eye [145]. A very recent study presented a rat model that developed reproducible quiescent CNV (without signs of exudation) by subretinal injection of an adeno-associated virus-VEGFA165 vector [146]. The model may become useful to investigate the long-term effects of new drugs targeting CNV under defined conditions [146]. Also very recently, Wang et al. reported that IQ protein motif-containing GTPase activating protein 1 (IQGAP1), a scaffold protein with a Rac1 binding domain, regulated VEGF activation by binding to Rac1GTP in choroidal endothelial cells, activating their migration [147]. Vascular permeability and angiogenesis in the retina are initiated by the VEGF-A/VEGFR2 signaling pathway, which is responsible for stimulating proliferation and migration of vascular endothelial cells [148]. Clinical application of anti-VEGF agents has improved the management of neovascular AMD, but requires repeated intraocular injections, which is only effective in approximately 40% of eyes [149,150]. Targets downstream of VEGF, such as NOX or Rac1, may provide more effective and safer therapies for neovascular AMD [149].

In addition to the VEGF mechanism, vascular endothelial dysfunction is considered a crucial event in the development and progression of choroidal vascular dysfunction [151]. Nitric oxide synthases (NOSs) are a family of enzymes that catalyze nitric oxide (NO) production from L-arginine, and are classified into three isoforms: endothelial NOS (eNOS), neuronal NOS (nNOS), and inducible NOS (iNOS) [152]. The eNOS isoform plays a critical role in maintaining the physiological functions of the vascular endothelium [153]. Based on experiments in mice lacking individual *NOS* genes, it was demonstrated that eNOS mediated endothelium-dependent vasodilation in retinal arterioles and ophthalmic arteries [154,155]. In retinal arterioles, the lack of eNOS was partially compensated by nNOS and cyclooxygenase-2 metabolites, whereas in ophthalmic arteries a compensation of endothelium-dependent vasodilator responses was mediated via endothelium-derived hyperpolarizing factors (EDHFs) [133,135,136]. NO generated by eNOS is not only a mediator of vasodilation, but is also a regulator of various other vascular functions [156]. For example, physiological levels of NO can dilate a blood vessel by relaxation of vascular smooth muscle cells, inhibiting vascular smooth muscle cell proliferation, and regulating angiogenesis and vascular permeability [157,158]. The choroidal vasculature is well known to receive autonomic innervation via sympathetic and parasympathetic nerves [159]. The nNOS isoform is present in perivascular nerve fibers and constitutes a major source of arteriolar NO [160]. Bhutto et al. reported that expression of eNOS and nNOS was significantly downregulated in the eyes of patients with AMD [161]. The authors of the study suggested that the decrease in eNOS and nNOS expression might have resulted in reduced NO production, which might be a reason for hemodynamic changes in CNV [161].

However, the production of NO is not always beneficial, since excessive amounts of NO can have detrimental effects on cells and tissues [156]. For example, NO can be an important stimulator of CNV. It has been suggested that iNOS produces large amounts of NO, which may promote CNV formation [162]. This indicates that the NOS isoform and the quantities of NO need to be considered in the pathophysiology of AMD [162]. One study suggested that the non-isoform-selective NOS inhibitor, N(G)-monomethyl-L-arginine (L-NMMA), may protect from CNV formation [162]. Based on a study in NOS gene knockout mice, Ando et al. suggested that blockade of nNOS and iNOS reduced CNV formation [163]. In addition, a reduction of CNV formation was observed in cell cultures by downregulation of the iNOS/NO/VEGF signaling pathway [164]. Furthermore, the interplay between NO and ROS can lead to the generation of peroxynitrite (ONOO^−^), a reactive ion, which compromises vascular endothelial function [165].

### 4.3. ROS and Choroidal Vascular Dysfunction

There is some evidence that ROS and vascular dysfunction may contribute together to the pathology of neovascular AMD [149]. NOX have been demonstrated as one connection between VEGF and ROS in human choroidal endothelial cells [149]. The family of NOX consists of 7 catalytic homologues: NOX1, NOX2, NOX3, NOX4, NOX5, dual oxidase (Duox) 1 and Duox2, which are differentially expressed in tissues and cells [166]. NOX1, NOX2, and NOX4 were reported to be expressed in choroidal vascular endothelial cells [167]. In addition, NOX have emerged as major sources of ROS in the vasculature [166]. NOX-generated ROS can function as signaling molecules promoting endothelial cell proliferation, migration and tube formation [166,167]. Other studies demonstrated that NOX2-derived ROS activate the transcription factors NF-κB and activator protein 1 (AP-1), and play a major role in increasing expression of the intracellular adhesion molecule (ICAM)-1 and VEGF, which is associated with vascular hyperpermeability and retinal neovascularization [168,169,170]. Moreover, NOX4-derived ROS generation is essential for hypoxia-inducible factor 1-alpha (HIF-1α)-dependent VEGF expression, which was linked to cell proliferation and migration in vascular smooth muscle cells [171]. NOX4 in vascular endothelial cells contributed to VEGF-induced pathologic angiogenesis and neovascularization through VEGF/VEGFR2 signaling and the extracellular signal-regulated kinase (ERK) pathway [167,172]. Laboratory evidence provides basic support for ROS-stimulating VEGF in the pathophysiology of AMD [173]. In cultured human RPE exposed to H_2_O_2_, ROS increased expression of a cell-associated splice variant of VEGF-A, VEGF189. A coculture of such treated RPE with choroidal endothelial cells was shown to facilitate choroidal endothelial cells to migrate across the RPE, which is considered a critical step in the development of vision-threatening neovascular AMD, by activating VEGFR2 and Rac1 that appears independent of the phosphoinositol 3-kinase (PI-3K) signaling pathway [174]. Rac1 is a subunit of NOX and plays an important role in directed endothelial cell motility [174]. The investigators proposed that downregulation of the NOX subunit, p22phox, an integral component of the NOX multi-component enzyme complex, by siRNA in RPE cells, decreased the generation of ROS, reduced VEGF production, and therefore led to a delay in the pathogenesis of CNV [175]. Soluble VEGF in turn can activate Rac1 upstream from NOX in human choroidal endothelial cells and further increase generation of ROS, contributing to choroidal neovascularization [149]. VEGF stimulates ROS production by activation of NOX (e.g., NOX2), and ROS are involved in VEGFR2-mediated signaling linked to endothelial cell migration and proliferation and tube formation in angiogenesis of choroidal neovascularization [149,176].

Under physiological conditions, eNOS generates the vasoprotective molecule NO in the vascular endothelium, which plays a pivotal role in regulating vascular tone as well as vascular homeostasis [156]. Vascular NO dilates vascular tone by stimulating the soluble guanylyl cyclase and inhibiting leukocyte and platelet aggregation and adhesion to the vascular wall [177]. On the one hand, a growing amount of evidence suggests that ROS can uncouple the eNOS enzyme from the oxidation of L-arginine, resulting in reduced NO generation [178]. Apparently, uncoupling of eNOS may require NOX, a primary superoxide anion source [178]. Alternatively, high levels of ROS may reduce vascular bioavailability of NO by direct reaction with NO, thereby generating a deleterious free radical called ONOO^−^ [179]. On the other hand, ROS instead of NO can be produced by uncoupling eNOS due to a deficiency in the enzyme cofactor tetrahydrobiopterin (BH4) or L-arginine [180]. Additionally, ONOO^−^ can uncouple eNOS by oxidizing BH4 to BH3 and further to BH2, which can compete with BH4 for binding to eNOS’s oxygenase domain, finally resulting in eNOS uncoupling [181]. In retinal tissue, ONOO^−^ can contribute to the increase in permeability of the microvascular endothelium, leading to vascular leakage [182]. Figure 3 describes the interplay between ROS and CNV formation.

## 5. Therapy Strategies in AMD Targeting the Vasculature and Oxidative Stress

### 5.1. Current Therapies

According to the multifactorial pathogenesis of AMD, various treatment options have been considered in clinical practice. Oral supplements may be beneficial in providing antioxidant protection against oxidative damage [183]. According to the Age-Related Eye Disease Study 2 (AREDS 2), antioxidant compounds, such as vitamin C, E, beta-carotene, lutein, zeaxanthin, and zinc, may help to protect the macula against loss of vision [184]. Curcumin, a naturally occurring substance present in turmeric, appears to exert antioxidant effects through inhibition of the NF-κB pathway [185]. Moreover, two decades of prospective follow-up showed that a higher intake of bioavailable lutein and zeaxanthin is related to a long-term reduced risk of advanced AMD [186].

Moreover, both in vitro and in vivo studies suggest that statins exhibit a potential diverse protective role, which reaches beyond lipid-related therapeutic properties [187,188,189]. For example, the findings of Wagner et al. suggest that the three 3-hydroxy-3-methylglutaryl coenzyme A (HMG-CoA) reductase inhibitors, atorvastatin, pravastatin, and cerivastatin affect NO/O_2_^−^ balance by upregulating eNOS expression and preventing the isoprenylation of p21 Ras-related C3 botulinum toxin substrate (Rac) to inhibit endothelial O_2_^−^ formation [189]. Notably, there are several pathophysiological mechanisms of statins in the treatment of AMD, which include anti-oxidative stress, direct anti-inflammatory action, anti-angiogenic effects, LDL and peroxidized lipids downregulation, and vascular endothelial function enhancement [190,191,192,193]. Additionally, statins were shown to downregulate VEGF expression in RPE cells in an experimental AMD model by downregulating the receptor for advanced glycation end-products [194]. Therefore, it is worth considering that statins may become a potential treatment for AMD.

High doses of statins, such as rosuvastatin calcium (Crestor, AstraZeneca), simvastatin (Zocor, Merck), and atorvastatin (Lipitor, Pfizer), were found to potentially clear the lipid drusen in patients with AMD studied in an early-stage clinical trial [195]. In the literature, there is a controversy regarding the association between statin intake and AMD incidence. In 2015, Al-Holou et al. reported that statin use was not significantly associated with the progression to late AMD and that there was a lack of evidence for a beneficial effect on slowing AMD progression in the AREDS2 participants [196]. In 2016, Gehlbach et al. indicated that data from available randomized controlled trials was insufficient to demonstrate that statins prevent the onset or progression of AMD [197]. However, other studies suggested that controlling serum cholesterol levels may be helpful in maintaining vision in AMD [198,199]. According to a population-based cohort study, Tan et al. demonstrated that statin use was not associated with early AMD incidence, but was protective for indistinct soft drusen after a 10-year follow-up of the initial baseline population [200]. A meta-analysis showed that statin use was protective for AMD, but the group of statins studied or the dosage administered was not specified [201]. Based on the blood-aqueous barrier and the poor aqueous solubility of statins, oral administration of statins leads to limited ocular concentrations. Nano-sized drug delivery systems may become useful in enhancing the therapeutic potential of statins [202,203]. Just recently, Yadav et al. investigated for the first time the effects of topically applied atorvastatin (ATS; representative statin) loaded into solid lipid nanoparticles as self-administrable eye drops [204]. The authors proposed that topical application of statins as eye drops will achieve high ocular concentration resulting in a consistent therapeutic effect [204].

Brolucizumab (Beovu^®^) is a VEGF inhibitor being developed by Novartis, which received its first approval on 8 October 2019 in the United States for the treatment of wet AMD [205]. Brolucizumab inhibits the three major VEGF-A isoforms (VEGF110, VEGF121, and VEGF165), suppressing endothelial cell proliferation, CNV formation and vascular permeability [138]. Apart from brolucizumab, there are three other drugs that have been previously approved by the United States Food and Drug Administration (FDA) for the treatment of exudative AMD by injections into the vitreous cavity: ranibizumab (Lucentis^®^, Lucentis, Genentech Inc., San Francisco, CA, USA), aflibercept (Eylea^®^, Regeneron Pharmaceuticals Inc., Tarrytown, NY, USA), and off-label bevacizumab (Avastin^®^, Genentech Inc., San Francisco, CA, USA) [206].

### 5.2. Novel Gene Therapies

Gene therapy is emerging as an innovative direction to treat genetic diseases. The basic mechanism of gene therapy involves implanting genetic material into the eye in order to correct a dysfunctional gene or to code for a therapeutic protein minimizing systemic absorption of gene vectors [207,208]. Since AMD is a complex multifactorial disease, in which genetic factors play a critical role, gene therapy is offering a novel tool and may represent the future of AMD treatment especially on the common and currently untreatable GA [209].

Complement activation pathways with genetic variants of complement C3 are strongly associated with the progression of AMD, as complement C3 is a point of convergence for 3 activation pathways (classical, alternative and lectin pathway) [210]. Complement C3 and its activation products (C3a, C3c and C3d) have been found in basal laminar and linear deposits as well as in surgically removed CNV tissue, which may imply that dysregulation of complement C3 activation plays a particularly significant role in the pathogenesis of AMD [211,212]. In addition, Nozaki et al. reported that genetic ablation of receptors for C3a downregulates leukocyte recruitment, VEGF expression, and CNV formation in a model of neovascular AMD. Moreover, pharmacological blockade of C3a receptors was also shown to reduce CNV [213]. Trakkidas et al. reported that exposure to H_2_O_2_ stimulated accumulation of complement proteins C3 and CFH in ARPE-19 cells, leading to endogenous complement-dependent angiogenic and inflammatory responses [214]. Furthermore, an in vitro study revealed that C3a and C5a activate ROS production in human eosinophils, and that blockade of the C3a and C5a receptor inhibits ROS production [215]. Therefore, given the importance of complement C3 in CNV formation and ROS production, inhibition of complement C3 is considered to be a potential treatment for AMD [216]. Liao et al. conducted an 18-month Phase II study in the United States to evaluate the safety and efficacy of intravitreally administered pegcetacoplan, a pegylated complement C3 inhibitor peptide, in patients suffering from GA [216]. This study reported that intravitreal injections of pegcetacoplan 15 mg monthly or every other month can significantly slow down the progression of GA [216]. Another complement pathway inhibitor, GT005, is now in a human Phase I/II trial to evaluate the safety and efficacy of GT005 as a single subretinal injection in GA patients. This clinical trial will be completed in February 2025 [217].

Although therapies targeting VEGF in AMD significantly improve vision, anti-VEGF treatments with frequent intravitreal injections are inconvenient and may increase the risk of endophthalmitis [218]. Therefore, intraocular gene delivery of VEGF antagonists may become an attractive treatment alternative, which would avoid the need for frequent intravitreal injections [219]. Honda et al. constructed an adenovirus expressing an entire ectodomain of the human VEGF receptor/fms-like tyrosine kinase-1 (Flt-1) fused to the Fc portion of human IgG (Adflt-ExR) and reported on a reduction of fibroblast proliferation and inflammatory cell infiltration in the photocoagulation spot of Adflt-ExR-treated rats [220]. Gene therapy with subretinal injection of adeno-associated virus serotype 2 (AAV2) carrying the soluble form of the Flt-1 receptor (AAV2.sFlt-1), encoding the secreted form of Flt-1, prevented the development of CNV in many studies [220,221,222,223,224]. These gene therapy studies show AAV2.sFlt-1 via an intravitreal injection express a potent anti-VEGF molecule, resulting in reduction of CNV development, with well-tolerated and capable of long-term expression effects [225]. The first administered gene therapy approved by the FDA in 2017 was Luxturna (voretigene neparvovec-rzyl) for treatment of biallelic RPE65-mediated inherited retinal diseases by using a recombinant adeno-associated virus (AAV) [226]. With the success of the first gene therapy approved by the FDA, the number of ongoing gene therapy studies is increasing. In 2020, two gene therapies are in early trials for treatment of AMD: ADVM-022 (Adverum Biotechnologies) and RGX-314 (Regenxbio) are delivered subretinally and intravitreally, respectively [227]. Intravitreous injection of AAV2-sFLT01 in patients with CNV seemed to be safe and well-tolerated in all doses of a Phase I open-label trial [228]. Additionally, intravitreal HMR59 (AAVCAGsCD59) for the gene treatment of AMD via blocking the complement system at the membrane attack complex (MAC) is in a Phase II dry AMD clinical trial [229]. Furthermore, intravitreal injection of PF-04523655, a synthetic 19-mer siRNA, targeting the RTP801 gene, seems to be safe in clinical trials for the treatment of neovascular AMD [230]. Gene therapy with a single subretinal or intravitreal injection may offer longer-lasting effects in visual gains in both dry and wet AMD, hence reducing or eliminating the need for frequent intravitreal applications of VEGF inhibitors [208].

## 6. Conclusions

We have highlighted the important mechanisms of oxidative damage, dysregulated lipid metabolism, choroidal vascular dysfunction and genetic factors in the pathogenesis of AMD, and the potential interrelations underlying these mechanisms. Oxidative damage is playing a central role in choroidal vascular dysfunction, which in turn triggers excess production of ROS in AMD. In addition, some gene polymorphisms, e.g., of the *CFH* and the *APOE* genes, were shown to strongly increase AMD susceptibility. Future well-tolerated treatments aimed at protecting against oxidative damage and excess VEGF production will be part of a multifaceted approach, encompassing a deeper understanding of the processes underlying AMD. Of note, gene therapy is offering a novel therapeutic approach and may represent the future of AMD treatment.

## Figures and Tables

**Figure 1 ijms-22-01296-f001:**
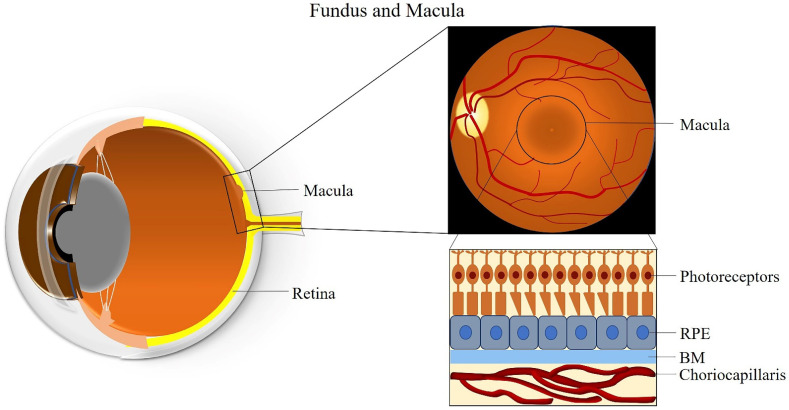
Anatomy of the fundus and macula. Abbreviations: RPE: retinal pigment epithelium; BM: Bruch’s membrane.

**Figure 2 ijms-22-01296-f002:**
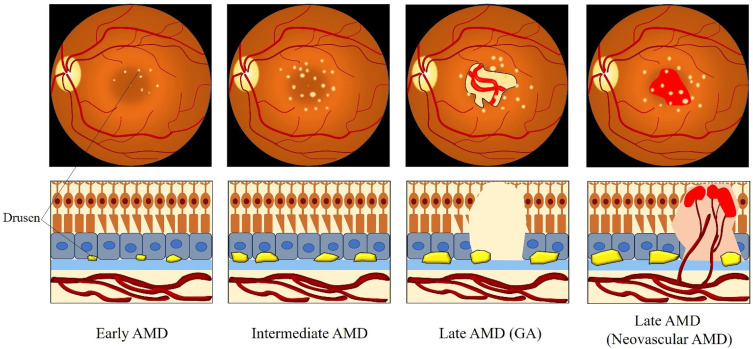
Clinical manifestation and pathology of AMD from the early to late stage. Medium-sized drusen found in early AMD. Intermediate AMD shows the presence of large drusen. Late AMD is classified into GA and neovascular AMD. GA is defined by the deterioration of the RPE, photoreceptor layer, and choroidal capillaries in the macula. The invasion of abnormal fragile choroidal blood vessels growing from the choroid into the retina in neovascular AMD, with blood and fluid leakage. Abbreviations: AMD: age-related macular degeneration; GA: geographic atrophy.

**Figure 3 ijms-22-01296-f003:**
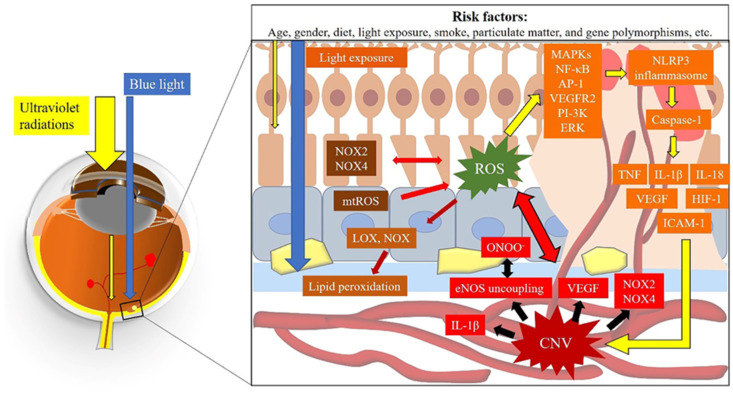
The interplay between ROS and CNV formation. Abbreviations: CNV: choroidal neovascularization; ROS: reactive oxygen species; mtROS: mitochondrial ROS; NOX: nicotinamide adenine dinucleotide phosphate oxidase; LOX: lysyl oxidase; eNOS: endothelial nitric oxide synthases; ONOO^−^: peroxynitrite; MAPKs: mitogen-activated protein kinases; NF-κB: nuclear factor-κB; VEGF: vascular endothelial growth factor; VEGFR2: vascular endothelial growth factor receptor 2; ERK: extracellular signal-regulated kinase; PI-3K: phosphoinositol 3-kinase; AP-1: activator protein 1; NLRP3: nod-like receptor family pyrin domain containing 3; TNF: tumor necrosis factor; IL-1β: interleukin-1β; IL-18: interleukin-18; HIF-1: hypoxia-inducible factor 1; ICAM-1: intercellular adhesion molecule-1.

**Table 1 ijms-22-01296-t001:** The Beckman clinical classification of age-related macular degeneration (AMD) [41].

Classification	Clinical Manifestation
No AMD	No drusen and no RPE abnormalities
Normal aging changes	Drusen ≤ 63 μm and no RPE abnormalities
Early AMD	Drusen > 63 μm and ≤125 μm and no RPE abnormalities
Intermediate AMD	Drusen > 125 μm and/or RPE abnormalities
Late AMD	GA and/or neovascular AMD

Abbreviations: AMD: age-related macular degeneration; GA: geographic atrophy; RPE: retinal pigment epithelium.

## Data Availability

Not applicable.

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
