# Peer review of "Age-Related Macular Degeneration: Role of Oxidative Stress and Blood Vessels"

_ijms, 2021, doi:10.3390/ijms22031296_

Round 1

Reviewer 1 Report

This is a quite good review, covering the related area. The authors need to check correct references cited, such as Line 285 reverse cholesterol transport (RCT) in the RPE, at least three recent publications for RCT in the RPE. 

Minor comments

Line 55 exposure to sunlight exposure

Figure 1, Insert for macula area, but the whole fundus was shown.

Author Response

Dear Reviewer,

We thank you for the valuable comments and suggestions. Please find our responses attached. 

Sincerely yours,

Yue Ruan

Reviewer 2 Report

This article meets all the requirements for review-type articles, and it is undoubtedly worth publishing. I do believe that this work will contribute towards the understanding of AMD disease. The introduction provides sufficient background. The article is written in an appropriate way. The data and figures are presented appropriately.

Minor spell check is required.

15 - “vascular function in in more detail“.

There are few abbreviations not explained.

195 - ARPE-19.

210 - A2E.

265 - ATP.

185 - DNA.

Figure 3 is missing VEGF abbreviation.

Author Response

(The authors gave the same response as above.)

Reviewer 3 Report

The impression of the manuscript is ambiguous. On the one hand, the topic of the review is relevant, on the other hand, according to the database, at least 30 reviews with the keywords "oxidative stress and AMD" were published in 2020 alone. That is, the topic of the manuscript is well and in detail covered in the literature, including a fresh, solid review by the same authors, published in the MDPI journal (Antioxidants 2020, 9(8),761; https://doi.org/10.3390/antiox9080761 Oxidative Stress and Vascular Dysfunction in the Retina: Therapeutic Strategies). Under these conditions, it is difficult for reviewers to do without comparisons with similar review articles. Realizing that it is difficult to write a high-quality review, nevertheless, I must admit that this manuscript loses to the rest of the review articles, as well as the previous review article by the authors, by several criteria.

Reading the review article from Q1 journal, the reviewer expects to see the author's position on the issue and a critical analysis of the results, not just a dry listing of facts. However, I have failed to understand the idea and purpose of this manuscript. It seems that the authors made an attempt to repeat the previous review by narrowing the focus on AMD. What provisions, what conclusions, what idea should the reader take from the review? The position and the rationale for own interpretation, should sound in a good review and link the sections of the review like a red thread. Authors should reshape the narrative: it is not necessary in a thematic issue of an excessive description of the anatomy of the retina, a digression into history and numerical data on the structure of morbidity. However, it is worthwhile to describe in more detail the known molecular mechanisms on the basis of modern literature data, how they can lead to oxidative stress and vascular dysfunction in AMD, without avoiding a causal relationship. How does section 5 relate to the topic of the article? It is not clear what is the connection between the new vector systems and the role of oxidative stress and vascular dysfunction in AMD ... What does the long paragraph about statins have to do with it? But the important points are said in passing. For example, the review lacks data on the role of mitochondrial damage, transcription factors (Nfe2l2/Are, PGc-1), and the contribution of cell death pathways in connection with vascular dysfunction and ROS formation. There is very little information about oxidative stress animal models (PGC1a KO, Sod2 KO, Nrf2 KO, PPARb KO et al).

I want to give the authors a chance to rewrite the review, despite the large amount of work on this topic with clearer analytics and stronger analysis of mechanisms.

Author Response

(The authors gave the same response as above.)

Reviewer 4 Report

This is a well structured and informative review about age-related macular degeneration and the role of reactive oxygen species and choridal vascular functions. 

Here are some minor points I would add to the manuscript:

- line 322 - 337: It is not only CFH which is a common genetic risk factor for AMD, but also the complement components CFHR3/CFHR1, C3, CFI, C2/CFB (e.g. Fritsche et al. 2016 (Nature Genetics) doi:10.1038/ng.3448). Please describe and mention this part more precisely with up to date publications.

- line 555 - 577: In the part "5. Therapy strategies in AMD" i would add a paragraph for relevant and newest clinical trials, especially for dry AMD (e.g. Liao et al. 2019 (Ophthalmology) doi:10.1016/j.ophtha.2019.07.011). 

Author Response

(The authors gave the same response as above.)

Round 2

Reviewer 3 Report

 accept